# Antimicrobial Action Mechanisms of Natural Compounds Isolated from Endophytic Microorganisms

**DOI:** 10.3390/antibiotics13030271

**Published:** 2024-03-18

**Authors:** Farkhod Eshboev, Nilufar Mamadalieva, Pavel A. Nazarov, Hidayat Hussain, Vladimir Katanaev, Dilfuza Egamberdieva, Shakhnoz Azimova

**Affiliations:** 1S. Yu. Yunusov Institute of the Chemistry of Plant Substances, Academy of Sciences of Uzbekistan, Mirzo Ulugbek Str. 77, Tashkent 100170, Uzbekistan; nmamadalieva@yahoo.com (N.M.); genlab_icps@yahoo.com (S.A.); 2School of Chemical Engineering, New Uzbekistan University, Movarounnahr Street 1, Mirzo Ulugbek District, Tashkent 100000, Uzbekistan; 3Institute of Fundamental and Applied Research, National Research University TIIAME, 39 Kori Niyoziy Str., Tashkent 100000, Uzbekistan; egamberdievad@gmail.com; 4Faculty of Biology, National University of Uzbekistan, Tashkent 100174, Uzbekistan; 5A.N. Belozersky Institute of Physico-Chemical Biology, Lomonosov Moscow State University, 1/40 Leninskie Gory, Moscow 119991, Russia; nazarovpa@gmail.com; 6Leibniz Institute of Plant Biochemistry, Department of Bioorganic Chemistry, Weinberg 3, D-06120 Halle, Germany; hussainchem3@gmail.com; 7Translational Research Center in Oncohaematology, Department of Cell Physiology and Metabolism, Faculty of Medicine, University of Geneva, 1211 Geneva, Switzerland; vladimir.katanaev@unige.ch; 8School of Medicine and Life Sciences, Far Eastern Federal University, Vladivostok 690090, Russia

**Keywords:** antibiotics, natural compounds, endophytes, mechanisms of action, antibacterial resistance, biosynthetic gene clusters

## Abstract

Infectious diseases are a significant challenge to global healthcare, especially in the face of increasing antibiotic resistance. This urgent issue requires the continuous exploration and development of new antimicrobial drugs. In this regard, the secondary metabolites derived from endophytic microorganisms stand out as promising sources for finding antimicrobials. Endophytic microorganisms, residing within the internal tissues of plants, have demonstrated the capacity to produce diverse bioactive compounds with substantial pharmacological potential. Therefore, numerous new antimicrobial compounds have been isolated from endophytes, particularly from endophytic fungi and actinomycetes. However, only a limited number of these compounds have been subjected to comprehensive studies regarding their mechanisms of action against bacterial cells. Furthermore, the investigation of their effects on antibiotic-resistant bacteria and the identification of biosynthetic gene clusters responsible for synthesizing these secondary metabolites have been conducted for only a subset of these promising compounds. Through a comprehensive analysis of current research findings, this review describes the mechanisms of action of antimicrobial drugs and secondary metabolites isolated from endophytes, antibacterial activities of the natural compounds derived from endophytes against antibiotic-resistant bacteria, and biosynthetic gene clusters of endophytic fungi responsible for the synthesis of bioactive secondary metabolites.

## 1. Introduction

Infectious diseases remain one of the leading global health concerns [1,2,3]. In 2019, 13.66 million people died globally because of infection-related causes. Most of these infection-related deaths (4.95 million) are associated with antibiotic-resistant pathogens [4]. It is predicted that in 2050, the number of deaths will significantly increase compared to the present because of antibiotic-resistant bacteria [5]. Therefore, finding novel antibiotics is considered an urgent task for scientists. However, the search for new antibiotics among cultivated organisms is complicated by the fact that more than approximately one million new actinomycetes must be considered to find just one new antibiotic, which greatly increases the cost for obtaining antibiotics [6]. High-throughput screening and fermentation methods, mining genomes for cryptic pathways, and combinatorial biosynthesis to generate new secondary metabolites related to existing pharmacophores provide some progress in the search for antibiotics among actinomycetes [7]. In addition, alternative approaches to the search for new antibacterial compounds are actively used, for example, searching among uncultured microorganisms [8], in silico screening of libraries of chemical compounds using deep-machine-learning methods [9], and searching among the microbiota of insects [10] and animals [11]. Another promising source of new antibiotics is endophytic microorganisms. Indeed, endophytes are a valuable source of metabolites because of their great species diversity and adaptability to a variety of environments [12,13]. Additionally, secondary metabolites of endophytes demonstrate a wide range of biological activities, including antibacterial, antifungal, anticancer, antiviral, antioxidant, anti-inflammatory, immunomodulatory, and hepatoprotective effects [14,15,16]. In recent years, interest in the secondary metabolites of endophytes has significantly increased. This interest has led to the successful development and practical implementation of several pharmaceutical drugs based on the secondary metabolites of endophytes. Some examples include the anticancer drug paclitaxel, the antibacterial and hepatoprotective compound piperine, the antimalarial agent quinine, and the topoisomerase enzyme inhibitor camptothecin [13].

Endophytes are a diverse group of microorganisms that reside within internal tissues of plants, forming symbiotic relationships without causing apparent harm to their hosts. They constitute the plant endosphere (which includes the endorhizosphere and endophyllosphere) and are an integral part of the plant microbiome along with the phyllosphere and rhizosphere. These microorganisms, including fungi, bacteria, and archaea [17], have long since attracted the attention of biologists and pharmacologists because of their diverse bioactivities and potential pharmacological applications [18,19,20]. Many natural compounds obtained from endophytes exhibit robust antimicrobial and anticancer properties [21]. Unfortunately, only a limited portion of these active substances have been thoroughly investigated regarding their mechanisms of action. Furthermore, the effects of endophyte-derived compounds with antibacterial activities on antibiotic-resistant bacteria have not been sufficiently studied. Another limitation is that the potential impacts of substances with anticancer properties on normal cells have not been adequately explored [22,23,24,25,26,27]. In many cases, these limitations arise from the difficulty in isolating pure active substances from endophytes and the low yields of the resultant compounds—issues that are further complicated by the fact that some of the compounds are synthesized only during a limited phase of the endophytes’ life cycle [28,29,30]. Therefore, it is important to identify biosynthetic gene clusters for biologically active secondary metabolites of endophytes [31]. With such identification, it would be possible to dissect the biosynthesis pathway of a given secondary metabolite and the main genes encoding the enzymes responsible for the biosynthesis. The cloning of genes and their expressions in other organisms suitable for large-scale metabolite production may allow the targeted and scaled biosynthesis of the desired bioactive substances in the future [32,33,34,35]. This review provides a comprehensive analysis of the mechanisms of the antibacterial action of endophyte-derived secondary metabolites, with a special focus on their impacts on antibiotic-resistant bacteria. We further present and discuss the studies dissecting the biosynthetic gene clusters controlling the biosynthesis of these metabolites from endophytic fungi.

## 2. Mechanisms of Action of Antimicrobial Drugs

Antibacterial compounds can be categorized into six primary groups based on their mechanisms of action. These mechanisms include the inhibition of cell wall synthesis, changes to the plasma membrane’s integrity, the disruption of the generation of cellular energy, damage to the synthesis of nucleic acids, the disruption of protein synthesis, and modulations to key metabolic pathways (Figure 1) [36,37,38].

The cell wall plays a crucial role in providing rigidity to bacterial cells. There are numerous classes of antibiotics that target the biosynthesis of cell wall components in bacteria, such as β-lactams, glycopeptides, and bacitracin [39]. The β-lactam ring-containing antibiotic penicillin inhibits the transpeptidase that catalyzes the cross-linking of peptidoglycan strands, which are major constituents of the cell wall [40]. Vancomycin, a glycopeptide antibiotic, inhibits cell wall formation by binding with the D-alanine–D-alanine motif of peptidoglycan precursors. Moreover, it inhibits transpeptidases and D,D-carboxypeptidases, catalyzing important biochemical reactions in cell walls [39]. Bacitracin inhibits the delivery of peptidoglycan precursor units to the cell membrane by inactivating the phospholipid carrier [41].

The cell membrane serves as a barrier for the uncontrolled diffusion of water, ions, and nutrients and, instead, mediates the controllable transport of the needed molecules [42]. Antimicrobial drugs specifically targeting the cytoplasmic membrane affect both Gram-positive and Gram-negative bacteria [43]. For example, polymyxins A–E disrupt the structure of membrane phospholipids and increase cellular permeability. Gram-negative bacteria are more sensitive to these antibiotics because of their high phospholipid contents both in the cytoplasm and outer membranes [44]. Unfortunately, membrane-targeting compounds are less selective as they can also damage human cell membranes [45].

Energy production through a network of metabolic processes is vital to any cell, including bacterial ones. Diverse cellular activities require energy to fuel them, either in the form of ATP or in the form of the membrane potential. An example of antibiotics that block ATP production is bedaquiline, an inhibitor of the membrane–bound F1Fo–ATP synthase [38]. An example of antibiotics that block the generation of the membrane potential is SkQ1, acting by the induction of protonophore-like futile cycles [46].

Antibiotics that inhibit nucleic acid synthesis are widely used in clinical settings. Their efficacy against various bacterial infections stems from their ability to disrupt essential processes of bacterial replication and transcription. Such antibiotics are classified into two main groups: those that target bacterial DNA and RNA synthesis by the inhibition of enzymes participating in these processes and those acting as nucleotide analogs incorporating into polynucleotide chains during DNA or RNA synthesis [47,48]. For example, fluoroquinolones (ciprofloxacin, levofloxacin, and moxifloxacin) inhibit the DNA gyrase and topoisomerase IV enzymes of bacteria [49,50,51,52]. Similarly, rifamycins (rifampin, rifabutin, and rifapentine) inhibit the bacterial RNA polymerase activity [53,54,55]. On the other side, antibiotics that incorporate into growing nucleic acids may have considerable side effect toxicities against human cells [56].

A popular class of antibiotics is represented by the drugs inhibiting bacterial protein synthesis, such as aminoglycosides, tetracyclines, macrolides, and chloramphenicol. Typically, antibiotics of this type target bacterial ribosomes. Because human 80S ribosomes and bacterial 70S ribosomes are structurally different, significant selectivity in the actions of the antibiotics can be achieved [57,58,59]. Aminoglycosides (streptomycin, gentamicin, and tobramycin) bind the 30S-subunit of the bacterial ribosome, causing the misreading of the mRNA during translation and the incorporation of incorrect amino acids into the growing polypeptide chain [60,61]. Tetracyclines (tetracycline, doxycycline, and minocycline) also act at the 30S-subunit of the bacterial ribosome and obstruct the binding of aminoacyl-tRNA to the A-site, preventing the addition of new amino acids to the growing peptide chain and inhibiting protein elongation [62,63]. Macrolides (erythromycin, azithromycin, and clarithromycin) interact with the 50S-subunit of the bacterial ribosome and interfere with the translocation of the peptidyl-tRNA from the A-site to the P-site of the ribosome [64,65].

Yet another class of antibiotics influences bacterial growth and survival by altering important metabolic processes. For example, trimethoprim and sulfonamides, representing two distinct classes of antibiotics, often work synergistically as a combination therapy to target bacterial infections. Both antibiotics interfere with the synthesis of tetrahydrofolate, a crucial compound involved in the production of nucleotides [66]. By targeting different steps in this pathway, trimethoprim inhibits dihydrofolate reductase, the enzyme responsible for the step of the tetrahydrofolate synthesis [67,68], while sulfonamides (sulfamethoxazole, sulfamethazine, and sulfadiazine) competitively inhibit dihydropteroate synthase thanks to their structural analogy to para-aminobenzoic acid, a precursor in the synthesis of dihydropteroate [69,70].

Although many antibiotics have been discovered to date, people around the globe continue to die from bacterial infections. Collaborative efforts of researchers, clinicians, and pharmaceutical companies are required to develop effective new antimicrobial drugs. New insights into the mechanisms of action of antibiotics and new sources to search for antibacterial substances are in need. In this regard, recent discoveries of antibacterial compounds originating from endophytic microbes may yield next-generation antibiotics. Desperately awaited, these active compounds may save human lives currently at peril because of bacterial infections, especially those resistant to the currently available treatments.

## 3. Antimicrobial Effects and Mechanisms of Action of Natural Compounds Isolated from Endophytes

Until now, a significant number of antimicrobial compounds have been isolated from endophytic microorganisms, demonstrating their potential as valuable sources of antimicrobial agents. However, despite the diverse array of antimicrobial compounds that have been identified, the mechanisms of action of only a small portion of them have been investigated [28]. In this section, we discuss these identified mechanisms of action and give the chemical structure of compounds isolated from endophytes (Figure 2).

Three prenylated indole alkaloids, asperglaucins A (**1**) and B (**2**) and neoechinulin F (**3**) have recently been isolated from *Aspergillus chevalieri* SQ-8, the endophyte obtained from *L. incana*. Compounds **1** and **2** demonstrated a significant antibacterial activity against *Pseudomonas syringae pv. actinidiae* and *Bacillus cereus*, with a minimum inhibitory concentration (MIC) of 6.25 μM. Scanning electron microscopy (SEM) revealed that **1** and **2** exhibited potential bacteriostatic effects by inducing structural changes in the external surfaces of *B. cereus* and *P. syringae pv. actinidiae* and causing cell membrane rupture or deformation [71].

(2E,5E)-Phenyltetradeca-2,5-dienoate (**4**), isolated from *Pseudomonas aeruginosa* strain UICC B-40, the endophyte obtained from *Neesia altissima*, exhibited an activity against *Staphylococcus aureus* ATCC 25923, with an MIC value of 62.5 μg/mL. According to SEM, the compound’s mechanism of action involves the breakdown of the bacterial cell wall leading to bacterial lysis [72].

Two new compounds, chetoseminudins F (**5**) and G (**6**), have been discovered alongside eleven previously known compounds from solid fermentation products of the endophytic fungus *Chaetomium* sp. SYP-F7950 obtained from *Panax notoginseng*. Among these compounds, chaetocochin C (**7**), chetomin A (**8**), and chetomin (**9**) demonstrated potent activities against *Staphylococcus aureus*, *Bacillus subtilis*, and *Enterococcus faecium*, as well as antifungal activity against *Candida albicans*. The MIC values for these activities ranged from 0.12 to 9.6 μg/mL. Molecular docking indicated that these compounds interact with the filamentous temperature-sensitive protein Z (FtsZ) of *Bacillus subtilis*. These findings suggest that the combined effects of FtsZ binding and the inhibition of cell division may be the mechanism of action against *B. subtilis* [73].

Two thiodiketopiperazine derivatives (**10** and **11**) have been obtained from *Phoma* sp. isolated from *Glycyrrhiza glabra*. These compounds possess activities against Gram-positive bacteria *Streptococcus aureus* and *S. pyogenes* with IC_50_ values of 5.8 μM and 3.1 μM (compound **10**) and 3.8 μM and 1.8 μM (compound **11**), respectively. Both compounds could inhibit biofilm formation by *S. aureus* and *S. pyogenes*. Furthermore, these compounds could inhibit transcription/translation processes and staphyloxanthin production in *S. aureus* [74].

Three new compounds, brasiliamide J-a (**12**), brasiliamide J-b (**13**), and peniciolidone (**14**), have been isolated from solid cultures of *Penicillium janthinellum* SYPF 7899 derived from *Panax notoginseng*. Additionally, eight known compounds have also been isolated. Among those, **12** displayed significant inhibitory activities against *Bacillus subtilis* and *Staphylococcus aureus*, with MIC values of 15 and 18 μg/mL, respectively. Compounds **13** and **14** showed medium inhibitory activities against *B. subtilis* (35 μg/mL and 50 μg/mL, respectively) and *S. aureus* (39 μg/mL and 60 μg/mL, respectively). The other compounds exhibited moderate to weak activities against the tested bacteria. In silico molecular docking studies have been conducted to gain further insights into the molecular interactions between the compounds and the target proteins. Compounds **12**–**14** demonstrated strong binding energies mediated by robust hydrogen bond and hydrophobic interactions with filamentous the temperature-sensitive protein Z (FtsZ) of *B. subtilis* and *S. aureus*. These findings highlight the potential of these compounds as FtsZ inhibitors, providing the basis for further exploration of their antimicrobial properties [75].

The flavonoid chlorflavonin (**15**), obtained from the endophytic fungus *Mucor irregularis*, isolated from *Moringa stenopetala*, exhibited an antibacterial activity against *M. tuberculosis*, with an MIC_90_ value of 1.56 μM. Molecular and docking techniques have revealed that **15** interacts with the acetohydroxyacid synthase catalytic subunit IlvB1, thereby inhibiting its activity [76].

Five prenylated benzaldehyde derivative compounds (dihydroauroglaucin (**16**), tetrahydroauroglaucin (**17**), 2-(3,6-dihydroxyhepta-1,4-dien-1-yl)-3,6-dihydroxy-5-(dimethylallyl)benzaldehyde (**18**), isotetrahydroauroglaucin (**19**), and flavoglaucin (**20**)) have been isolated for the first time from *Aspergillus amstelodami* (MK215708), an endophyte of *Ammi majus* L. Among these compounds, **16** showed high activities against *Escherichia coli*, *Streptococcus mutans*, and *Staphylococcus aureus*, with MICs of 1.95 µg/mL, 1.95 µg/mL, and 3.9 µg/mL, respectively. Additionally, **16** showed potent antibiofilm activity, with a minimum biofilm inhibitory concentration (MBIC) of 7.81 µg/mL against *S. aureus* and *E. coli* biofilms. It also displayed an MBIC of 15.63 µg/mL against *Streptococcus mutans* and *Candida albicans* biofilms and a moderate activity (MBIC = 31.25 µg/mL) against *Pseudomonas aeruginosa* biofilms [77].

Two natural bisantharaquinones, (+)-1,1′-bislunatin (**21**) and (+)-2,2′-epicytoskyrin A (**22**), isolated from endophytic fungi *Diaporthe* sp. GNBP-10, associated with *Uncaria gambir* Roxb, have shown promising anti-tubercular activities against the *Mycobacterium tuberculosis* strain H37Rv, with MIC values of 0.422 and 0.844 µM, respectively. Compounds **21** and **22** can also moderately inhibit biofilm formation in the *M. tuberculosis* model. Both compounds significantly reduce the number of *M. tuberculosis* infections within macrophages, resulting in a 2-fold log reduction. In silico docking reveals favorable affinities of **21** and **22** toward the enzyme pantothenate kinase, with glide scores of −7.481 kcal/mol and −8.427 kcal/mol, respectively [78].

Three compounds, xanthoascin (**23**), prenylterphenyllin B (**24**), and prenylcandidusin (**25**), have been isolated from *Aspergillus* sp. IFB-YXS, which is associated with the leaves of *Ginkgo biloba*. Compound **23** exhibits antibacterial activities against *Xanthomonas oryzae pv. oryzicola*, *Erwinia amylovora*, *Pseudomonas syringae pv*. *lachrymans*, and *Clavibacter michiganensis subsp*. *sepedonicus*, with MICs of 20, 10, 5.0, and 0.31 μg/mL, respectively. Compound **24** displays antibiotic activities with an MIC of 20 μg/mL against *X. oryzae pv. oryzicola, E. amylovora*, and *P. syringae pv. lachrymans*. Compound **25** shows effectiveness against *X. oryzae pv. oryzae* and *X. oryzae pv. oryzicola* with MICs of 10 and 20 μg/mL, respectively. It has been observed that **23** alters the permeability of the bacterial cytomembrane, causing the leakage of nucleic acids [79].

Two novel antibacterial terpene-polyketides, spiroaspertrione A (**26**) and aspermerodione (**27**), have been isolated from *Aspergillus* sp. TJ23, an endophytic fungus derived from the leaves of *Hypericum perforatum*. These compounds exhibited significant activities against methicillin-resistant *Staphylococcus aureus* (MRSA) with MICs of 4 µg/mL and 32 µg/mL, respectively. Further analysis has shown that these compounds inhibit the activity of penicillin-binding protein 2a, a key factor in MRSA resistance to β-lactam antibiotics. This inhibition results in a synergistic effect when **26** and **27** are combined with β-lactam antibiotics, such as oxacillin and piperacillin, enhancing their antibacterial activities against MRSA [80,81].

Septoreremophilane D (**28**) and (22E)-3β-hydroxy-26,27-bisnorcholesta-5,22-dien-24-one (**29**) have been isolated from the endophytic fungus *Septoria rudbeckiae* derived from *Karelinia caspia.* These compounds displayed potential antibacterial activities against *Pseudomonas syringae* pv. *Actinidiae* and *Bacillus cereus* (MIC = 6.25 μM for both). SEM has revealed that these substances cause alterations in the outer walls of bacterial cells [82].

In another study, ω-hydroxyemodin (**30**), isolated from *Penicillium restrictum* obtained from *Silybum marianum*, has noticeably inhibited quorum sensing in clinical isolates of methicillin-resistant *S. aureus* in both in vitro and in vivo models [83]. Quorum sensing is a communication mechanism used by bacteria to coordinate the expression of virulence factors and other aspects of their behavior. The inhibitory effects of **30** on quorum sensing highlight this compound as a potential therapeutic agent against methicillin-resistant *S. aureus* infections.

Two new compounds, penicimenolidyu A (**31**) and penicimenolidyu B (**32**), along with the known compound rasfonin (**33**), have been isolated from *Penicillium cataractarum* SYPF 7131, an endophyte of the plant *Ginkgo biloba*. Compound **33** demonstrated a significant antibacterial activity against *S. aureus*, with an MIC of 10 μg/mL. Compounds **31** and **32** exhibited moderate inhibitory activities against *S. aureus*, with MIC values of around 60 μg/mL. Molecular docking suggests that these compounds possess high binding energies, strong hydrogen bond interactions, and hydrophobic interactions with *S. aureus* FtsZ [84].

Penicillic acid (**34**), isolated from *Chaetomium elatum*, the endophytic fungi of the medicinal plant *Hyssopus officinalis*, has demonstrated strong antibacterial activities against Gram-positive (*B. subtilis* and *S. aureus*) and Gram-negative (*E. coli* and *P. aeruginosa*) bacteria with MIC values of 34–68 μg/mL. Moreover, **34** suppressed both biofilms and planktonic forms [85].

Four antimicrobial compounds, aspulvinones B’, H, R, and S (**35**–**38**, respectively), have been isolated from the fungus *Aspergillus flavus* KUFA1152, obtained from the marine sponge *Mycale* sp. Compounds **35**–**38** possess antibacterial activities against multidrug-resistant *Enterococcus faecalis* ATCC 29212 and *S. aureus* ATCC 29213 strains and inhibit the biofilm formation by these strains. The MIC values for these compounds ranged from 4 to 64 μg/mL [86].

A new phenolic compound, 4-(2,4,7-trioxa-bicyclo[4.1.0]heptan-3-yl) phenol (**39**), isolated from *Pestalotiopsis mangiferae*, obtained from *Mangifera indica* Linn, has shown potent antibacterial activities against *B. subtilis*, *Klebsiella pneumoniae* (MIC = 0.039 μg/mL), *E. coli*, *Micrococcus luteus* (MIC = 1.25 μg/mL), and *P. aeruginosa* (MIC = 5.0 μg/mL). The compound’s mode of action has been investigated using transmission electron microscopy, revealing that **39** caused morphological alterations in the cells of *E. coli*, *P. aeruginosa*, and *K. pneumoniae*. These alterations include the cytoplasm agglutination and formation of pores in cell wall membranes [87].

Two new aromatic butyrolactones, flavipesins A (**40**) and B (**41**), have been isolated from the endophytic fungus *Aspergillus flavipes* of the mangrove plant *Acanthus ilicifolius*. Compound **40** exhibits antibacterial activities against *S. aureus* and *B. subtilis* with MIC values of 8.0 μg/mL and 0.25 μg/mL, respectively, while **41** demonstrates both antibacterial and unique antibiofilm activities, penetrating the biofilm matrix and effectively killing live bacteria in mature *S. aureus* biofilms [88].

Three dimeric xanthones, phomoxanthone A (**42**), phomoxanthone B (**43**), and dicerandrol B (**44**), have been isolated from *Paecilomyces* sp. EJC01.1, the endophytic fungus of *Schnella splendens*. Among these compounds, **42** has revealed a significant antimicrobial activity specific for *B. subtilis* (MIC = 7.81 µg/mL). Molecular docking against S-ribosyl-homocysteine lyase (LuxS) suggests the interaction between **42** and two critical residues, His58 and Cys126, which play essential roles in the catalytic mechanism of LuxS in *B. subtilis*. Quantum studies utilizing density functional theory have further demonstrated a low gap value of 5.982 eV, indicating the high reactivity of this compound [89].

Five compounds—ergosterol (**45**), β-sitosterol (**46**), 5-pentadecylresorcinol (**47**), 5-hydroxymethyl-2-furancarboxylic acid (**48**), and succinimide (**49**)—have been isolated from *Aspergillus niger xj*. Among these compounds, **47** has exhibited antibacterial activity against *Ralstonia solanacearum* RS-2, with an MIC value of 15.56 μg/mL. SEM, assessment of the cell membrane permeability, and SDS-PAGE analysis have determined that the mechanism of action of **47** involves the inhibition of bacterial protein synthesis and intracellular metabolism [90].

Seven new indole diterpenoids, drechmerins A–G (**50–56**), have been isolated from the fermentation broth of *Drechmeria* sp. derived from *Panax notoginseng*’s roots. Among these compounds, **51** has displayed an antimicrobial activity against *C. albicans*, with an MIC value of 12.5 mg/mL. Molecular docking suggests that these compounds may interact with peptide deformylase [91]. Similarly, another indole diterpenoid, drechmerins I (**57**), isolated from the same fungus, shows antimicrobial activities against *B. subtilis* (MIC = 200 μg/mL). Molecular docking studies suggest that the interaction between **57** and peptide deformylase of *S. aureus* was strong [92].

Emodin (**58**), an anthraquinone compound, has been isolated from *Aspergillus awamori* WAIR120 and has demonstrated an activity against *E. faecalis* AHR7, with an MIC value of 125 μg/mL. Light microscopy analysis suggests that **58** induces morphogenic effects, such as the swelling and elongation of bacterial cells. Changes in the cell membrane and the submicroscopic structure of bacterial cells have further been observed by transmission electron microscopy. These observations suggest that **58** disrupts the integrity of the cell membrane, potentially leading to the leakage of the cellular contents and affecting the overall structure and function of bacterial cells [93].

The antibacterial-active fraction, containing the polyphenolic group (leucodelphinidin, dihydroquercetin, kaempferol, and quercetin) and one polyketide (patulin), derived from crude extracts of *Penicillium setosum* obtained from the roots of *Withania somnifera*, displays antibacterial activities against *E. coli* and *S. aureus*, with an MIC value of 8 μg/mL. Further, scanning electron micrographs provide visual evidence of morphological changes in the treated cells. In the case of *E. coli*, these changes include a reduction in the cell size and the appearance of bubbles and blisters on the cells’ surface. Similarly, in *S. aureus*, the treated cells exhibit open holes and deep craters on their surface, which can eventually lead to cellular rupture. Additionally, measuring the leakage of ions, such as Na^+^ and K^+^, from bacterial cells showed damage or permeabilization of the membrane [94].

Steffimycin B (**59**) is an anthracycline compound, isolated from *Streptomyces scabrisporus*, an endophyte found in the medicinal plant *Amphipterygium adstringens*. This compound has demonstrated a notable activity against *M. tuberculosis* H37Rv ATCC 27294, with an MIC_100_ value of 7.8 µg/mL. Interestingly, when tested against the rifampin-mono-resistant *M. tuberculosis* Mtb-209 pathogen strain, **59** exhibited an even higher activity, with an MIC_100_ value of 3.9 µg/mL. This suggests that **59** may have a distinct mechanism of action compared to rifampin, an antibiotic commonly used against tuberculosis. One proposed mechanism of action for **59** is its potential DNA-intercalating effect, implying that the compound may bind the DNA of *M. tuberculosis*, leading to an increase in the DNA-melting temperature. This observation suggests that **59** could disrupt DNA replication or transcription in bacteria, ultimately inhibiting their growth [95].

Five siderophores—SVK21 (**60**), bacillibactin C (**61**), bacillibactin B (**62**), tribenglthin A (**63**), and bacillibactin (**64**)—have been isolated from *B. subtilis* NPROOT3, a bacterial endophyte obtained from the halophyte *Salicornia brachiate*. Out of these compounds, **63** and **64** have demonstrated significant inhibitory activities against *M. smegmatis* MTCC6, with MIC values of 39 μM and 22 μM, respectively. Further studies conducted to determine the mode of action have shown that these compounds exhibit antibacterial activities by scavenging the iron ions in bacterial cells [96].

These studies of natural antimicrobial compounds derived from endophytes have provided valuable insights into their potentials as therapeutic agents against microbial infections [15,97]. Through rigorous research and screening processes, numerous endophyte-derived natural compounds have been identified and characterized for their antimicrobial activities, as exemplified above. The mechanisms of action underlying the antimicrobial activities of these natural compounds are multifaceted and vary depending on the specific compound and targeted microorganism. Some compounds exert their antimicrobial effects by disrupting cell membranes, leading to the leakage of cellular components and eventual cell death. Others interfere with key enzymatic processes vital for microbial survival, while some exhibit DNA intercalation or the inhibition of protein synthesis. Noteworthy, the mechanisms of action for a significant portion of these compounds remain largely unexplored [98]. Future studies should focus on employing advanced techniques and methodologies to unravel the intricate mechanisms of action underlying the antimicrobial activities of these endophyte-derived compounds. These future studies may employ OMICS approaches, structural biology techniques, and computational modeling to elucidate the interactions between these compounds and their microbial targets. In conclusion, the investigation of natural antimicrobial compounds derived from endophytes has great potential for the development of new antibiotics.

## 4. Effects of Natural Compounds Obtained from Endophytes on Antibiotic-Resistant Bacteria

Antimicrobial resistance (AMR) is a significant global health challenge. AMR can lead to prolonged illnesses, higher mortality rates, and increased healthcare costs. Microorganisms can develop various mechanisms to resist the effects of antibiotics, rendering these drugs less effective or completely ineffective in treating infections. This reduces the available treatment options for patients and can lead to increased morbidity and mortality rates. Furthermore, the development of new drugs to combat AMR has been relatively slow, leading to a limited number of novel drugs in the pharmaceutical pipeline [99,100].

Recent years have seen an increasing interest in the potential of natural compounds derived from endophytic microorganisms as sources of novel antimicrobial agents. The impacts of endophyte-derived substances on antibiotic-resistant bacteria have gained attention because of the potential of these compounds to provide effective solutions against multidrug-resistant pathogens [101,102]. Understanding the impacts of secondary metabolites derived from endophytes on antibiotic-resistant bacteria is important for several reasons. First, it can contribute to the discovery of new therapeutic options for the treatment of drug-resistant infections. Second, it may provide insights into novel mechanisms of action that can be exploited to overcome antibiotic resistance. Third, these compounds may have the potential to enhance the effectiveness of existing antibiotics when used in combination therapies [103]. In this section, we discuss the mechanisms for the development of resistance to antibacterial drugs by bacteria, as well as the current knowledge regarding the impacts of natural compounds obtained from endophytic microorganisms on antibiotic-resistant bacteria.

Multiple mechanisms of resistance to antibacterial drugs have been discovered by bacteria, including drug inactivation, modification of the antibiotic’s targets in bacteria, reduced intracellular drug accumulation due to decreased permeability and antibiotic efflux, formation of biofilms, overexpression of molecular targets that are affected by antibiotics, and utilization of alternative metabolic pathways [100,104,105].

The modification or inactivation of antimicrobial drugs is a commonly observed mechanism of resistance in numerous pathogenic bacteria. These bacteria possess genes that encode a variety of enzymes capable for modifying antibiotics. One such example is the hydrolysis of β-lactam antibiotics (penicillins, cephalosporins, and carbapenems) through the action of β-lactamases, which specifically hydrolyze the amide bond within the β-lactam ring [106,107,108,109]. Another common way that antibiotic resistance occurs is through the modification of antibiotic binding sites on their targets within bacterial cells. This mechanism involves altering the structure or function of the specific sites of the bacterial cell where antibiotics normally bind. For example, high-level vancomycin resistance in vancomycin-resistant enterococci is achieved through the modification of the pentapeptide stem of Lipid II. This modification involves the substitution of the terminal D-alanine residue with the isosteric D-lactate residue [110,111].

Many antibiotics, especially those targeting Gram-negative bacteria, face the challenge of the penetration of bacterial cells to exert their activities. This challenge becomes even more difficult when bacteria develop the mechanism of resistance that involves reducing the permeability of the cell envelope and increasing the activity of efflux pumps to decrease the intracellular accumulation of the antibiotics that should target intracellular processes. These mechanisms contribute to bacterial resistance by preventing antibiotics from reaching their intended targets at sufficient concentrations, making it more challenging to effectively treat bacterial infections [112,113]. For instance, in the *K. pneumoniae* strain (ST11) resistant to various antibiotics, such as β-lactams, sulfonamides, bacitracin, tetracycline, aminoglycosides, and chloramphenicol, five different families of multidrug resistance efflux pumps have been identified. These include the resistance nodulation division family, the ATP-binding cascade family, the small multidrug resistance family, the multidrug and toxic compound extrusion family, and the major facilitator family [114].

Biofilms play a significant role in bacterial antibiotic resistance. The biofilm structure consists of an extracellular polymeric matrix, a complex assembly of proteins, polysaccharides, and DNA. Biofilm-based bacterial antibiotic resistance occurs because of the low biofilm penetration of antibiotics; activities of matrix components (such as antibiotic-degrading enzymes and DNA); activation of specific stress response pathways in bacteria; slow reproduction of bacteria, which makes them less sensitive to certain antibiotics; and bacterial heterogeneity within biofilms that may include diverse antibiotic-resistant or persistent cells. Therefore, it is often observed that certain bacteria may be sensitive to specific antibiotics in their planktonic (free-floating) state but become resistant to the same antibiotics after biofilm formation. For example, the vancomycin resistance of *S. aureus* and tobramycin resistance of *P. aeruginosa* have been observed after the formation of their respective biofilms [115,116,117].

Another way to develop antibiotic resistance is the overexpression of the molecular targets of the antibiotics. Such overexpression can occur through various mechanisms, including genetic mutations, gene amplification, or increased transcription or translation rates [118]. For example, methicillin-resistant *S. aureus* has developed resistance to daptomycin by the overexpression of dltA, the molecular target of the drug [119].

By switching to alternative metabolic pathways, bacteria can bypass targets or processes inhibited by antibiotics, allowing them to survive and grow in the presence of these drugs. This adaptive response is a complex and dynamic process that involves metabolic rewiring, substrate utilization changes, and activation of redundant or parallel pathways [104]. For instance, the activation of an alternative transcriptional program allows for increased ATP production in antibiotic-resistant *S. aureus* [120].

Natural compounds isolated from endophytic microorganisms represent a promising alternative to plant-derived compounds, as well as to synthetic and semi-synthetic compounds, for the development of new antibiotics. Many substances isolated from endophytes demonstrate higher antimicrobial activities compared with those of synthetic and plant-derived compounds. In this section, we further discuss the antibacterial activities of recently isolated endophytic compounds (Figure 3) against antibiotic-resistant bacteria. For example, an endophytic fungus *Chaetomium globosum* 7s-1 isolated from *Rhapis cochinchinensis* (Lour.) Mart has yielded the following 10 natural compounds: xanthoquinodin B9 (**65**), xanthoquinodin A1 (**66**), xanthoquinodin A3 (**67**), chetomin (**9**), chaetocochin C (**68**), dethio-tetra(methylthio)chetomin (**69**), chrysophanol (**70**), emodin (**58**), alatinone (**71**), and ergosterol (**45**). Compounds **65**, **66**, and **67** have shown significant activities against *S. aureus* and methicillin-resistant *S. aureus*, with MIC values ranging from 0.87 to 1.75 μM. Moreover, **9** has demonstrated a strong activity against methicillin-resistant *S. aureus*, with an MIC value of 0.02 pM, which is remarkably lower than that of vancomycin (0.67 μM) [121].

L-tyrosine (**72**) has been isolated for the first time from *Rhizopus oryzae* AUMC14899 obtained from *Opuntia ficus-indica* (L.). Compound **72** showed strong antibacterial and antibiofilm activities against multidrug-resistant bacterial strains (*P. aeruginosa* PA-02, *P. aeruginosa* PA-09, *E. coli* EC-03, *Klebsiella pneumonia* KP-01, *S. aureus* SA-03, and *S. aureus* SA-04) isolated from burn wound infections, with MIC values from 6 to 20 µg/mL. In addition, **72** could strongly reduce biofilm formation and disrupt pre-formed biofilms [122].

Cyschalasins A (**73**) and B (**74**) have been isolated from the endophytic fungus *Aspergillus micronesiensis* derived from the root of *Phyllanthus glaucus*. These compounds exhibited antibacterial activities against methicillin-resistant *S. aureus*, with MIC_50_ values of 17.5 and 10.6 μg/mL, respectively [123].

Two new anthraquinones, 2′,6-dimethyl-7-methoxy-[2,3-b]furan-anthraquinone (**75**) and 1,7-dimethoxy-2′,6-dimethyl-[2,3-b]furan-anthraquinone (**76**), have been extracted from the endophytic fungus *Aspergillus versicolor* YNCA1208, obtained from cigar tobacco. At 1 μg/mL, **75** and **76** demonstrate strong antibacterial activities against the methicillin-resistant *S. aureus* strain ZR11, with inhibition zone diameters of 16.4 ± 2.2 and 18.5 ± 2.5 mm, respectively [124].

Eight new sesquiterpene eutyscoparins A–H (**77–84**) have been isolated from the EtOAc extract of the endophytic fungus *Eutypella scoparia* SCBG-8, obtained from leaves of *Leptospermum brachyandrum.* From these compounds, only **83** has demonstrated antibacterial activities against *S. aureus* and methicillin-resistant *S. aureus*, with an MIC value of 6.3 μg/mL [125].

Two new naphthalene derivatives, 5-methoxy-2-methyl-7-(3-methyl-2-oxobut-3-enyl)-1-naphthaldehyde (**85**) and 2-(hydroxymethyl)-5-methoxy-7-(3-methyl-2-oxobut-3-enyl)-1-naphthaldehyde (**86**), together with two known naphthalene derivatives parvinaphthol A (**87**) and pannorin B (**88**), have been isolated from fermentation products of the endophytic fungus *Phomopsis* sp., derived from the rhizome of *Paris polyphylla var. yunnanensis*. Compounds **85** and **86** display antibacterial activities against methicillin-resistant *S. aureus*, with inhibition zones of 14.5 ± 1.2 and 15.2 ± 1.3 mm, respectively [126].

The endophytic fungus *Pestalotia* sp., isolated from the leaves of *Heritiera fomes*, has yielded two compounds: oxysporone (**89**) and xylitol (**90**). These compounds demonstrate antibacterial activities against various strains of methicillin-resistant *S. aureus*, including ATCC 25923, SA-1199B, RN4220, XU212, EMRSA-15, and EMRSA-16. The MIC values of these compounds against methicillin-resistant *S. aureus* strains ranged from 32 to 128 μg/mL [127].

A total of nine natural compounds, seiricardine D (**91**); xylariterpenoid A (**92**); xylariterpenoid B (**93**); regiolone (**94**); 4-hydroxyphenethyl alcohol (**95**); (22E, 24R)5, 8-epidioxy-5α, 8α-ergosta-6,22E-dien-3ß-ol (**96**); (22E, 24R)5, 8-epidioxy-5α, 8α-ergosta-6,9(11), 22-trien-3ß-ol (**97**); ß-sitosterol (**98**); and stigmast-4-en-3-one (**99**), have been obtained from *Cytospora* sp. derived from the Chinese mangrove, *Ceriops tagal.* Among these compounds, only **96** has shown weak antibacterial activities against methicillin-resistant *S. aureus* GIM1.771, with an MIC value of ca. 230 µM [128].

Four new prenylxanthones, 14-hydroxyltajixanthone (**100**), 14-hydroxyltajixanthone hydrate (**101**), 14-hydroxyl-15-chlorotajixanthone hydrate (**102**), and epitajixanthone hydrate (**103**), have been isolated from the solid-substrate fermentation culture of *Emericella* sp. XL029 derived from the leaves of *P. notoginseng*. All the compounds showed a significant antibacterial activity against all the tested non-resistant bacteria, with MIC values ranging from 12.5 to 50 μg/mL. However, compounds **100** and **103** manifested moderate activities against drug-resistant *S. aureus*, with an MIC value of 50 μg/mL [129].

Seven compounds, namely, trichosetin (**104**), beauvericin (**105**), beauvericin A (**106**), enniatin B (**107**), enniatin H (**108**), enniatin I (**109**), and enniatin MK1688 (**110**), have been extracted from the endophytic fungus *Fusarium* sp. TP-G1 obtained from *Dendrobium officinale* Kimura et Migo. The MIC values of these compounds against *S. aureus* and methicillin-resistant *S. aureus* were comparable, ranging from 4 to 32 μg/mL [130].

Six new polyketides, aplojaveediins A–F (**111–116**), have been isolated from the endophytic fungus *Aplosporella javeedii* derived from *Orychophragmus violaceus*. Among these compounds, **111** and **116** demonstrate antibacterial properties against drug-sensitive *S. aureus* ATCC 29213, methicillin-resistant *S. aureus*, and vancomycin-intermediate-sensitive *S. aureus* ATCC 700699 strains [131].

Two new anthraquinones, 3-hydroxy-6-hydroxymethyl-2,5-dimethylanthraquinone (**117**) and 6-hydroxymethyl-3-methoxy-2,5-dimethylanthraquinone (**118**), have been isolated from fermentation products of the endophytic fungus *Phomopsis* sp. obtained from *Nicotiana tabacum L*. These compounds showed pronounced activities against methicillin-resistant *S. aureus*, with inhibition zones of around 14 mm [132].

A new alkaloid, preisomide (**119**), along with five known polyketides, namely, minimoidione B (**120**), preussochromone C (**121**), 7-hydroxy-2-(2-hydroxypropyl)-5-methylchromone (**122**), citreoisocoumarinol (**123**), and setosol (**124**), have been isolated from the endophytic fungus *Preussia isomera* obtained from *P. notoginseng*. Among these compounds, **124** has exhibited significant activity against three clinically relevant bacterial strains: multidrug-resistant *E. faecium*, methicillin-resistant *S. aureus*, and multidrug-resistant *E. faecalis*. For **124**, the MIC value against these bacterial strains was 25 μg/mL [133].

A culture of the endophytic fungus *Penicillium* sp. CPCC 400817, obtained from a mangrove plant collected from Dongzhai Harbor in Hainan Province, has yielded five alkaloid compounds, including one newly discovered compound named GKK1032C (**125**), as well as four known compounds: pyrrospirones E (**126**) and F (**127**), GKK1032B (**128**), and A2 (**129**). Compound **125** has demonstrated a potent activity against methicillin-resistant *S. aureus*, with an MIC value of 1.6 μg/mL. However, it has no activity against Gram-negative bacteria, such as *E. coli*, *P. aeruginosa*, and *Acinetobacter baumannii* [134].

Seven substances, including aureonitols A (**130**) and B (**131**), chaetoviridin G (**132**), chaetomugilin I (**123**), equisetin (**134**), and chaetoglobosins E (**135**) and F (**136**), have been obtained from a solid fermentation culture of the endophytic fungus *C. globosum* derived from the aerial parts of *Salvia miltiorrhiza*. Amidst these compounds, **134** displays significant activities against four multidrug-resistant bacteria: *E. faecalis*, *E. faecium*, *S. aureus*, and *S. epidermidis*, with MIC values of 3–6 μg/mL [135].

Three new terpene–polyketide hybrid meroterpenoids, emervaridones A–C (**137–139**), and two new polyketides, varioxiranediols A (**140**) and B (**141**), have been obtained from *Emericella* sp. TJ29 derived from *H. perforatum*. Compounds **137** and **140** are found to be effective against five drug-resistant microbial pathogens (methicillin-resistant *S. aureus*, *E. faecalis*, extended-spectrum β-lactamase-producing *E. coli* (ESBL-producing *E. coli*), *P. aeruginosa*, and *K. pneumoniae*), with MIC values in the microgram-per-milliliter range. Notably, the inhibitory effect of **137** against ESBL-producing *E. coli* is comparable to that of the clinically used antibiotic amikacin, with an MIC value of 2 μg/mL [136].

An extract of the endophytic fungus *Lecanicillium* sp. (BSNB-SG3.7 strain) derived from *Sandwithia guyanensis* has demonstrated a significant activity against methicillin-resistant *S. aureus* with an MIC value of 16 µg/mL. After the chemical investigation of the extract, five compounds have been isolated and identified as stephensiolides I (**142**), D (**143**), G (**144**), C (**145**), and stephensiolide F (**146**). The individual compounds showed activities against methicillin-resistant *S. aureus*, with MIC values ranging from 4 to 128 µg/mL [137].

Two new compounds, 1-(3-hydroxy-1-(hydroxymethyl)-2-methoxy-6-methylnaphthalen-7-yl) propan-2-one (**147**) and 1-(3-hydroxy-1-(hydroxymethyl)-6-methylnaphthalen-7-yl)propan-2-one (**148**), have been isolated from the endophytic fungus *Phomopsis fukushii* obtained from *N. tabacum.* These compounds show weak activities against methicillin-resistant *S. aureus*, with the diameter of the inhibition zone being 10–11 mm [138].

In another work, chetomin (**9**), isolated from *C. globosum* (HG423571) obtained from *Avena sativa*, has displayed a significant activity against multi(methicillin and oxacillin)-resistant *S. aureus* ATCC 700699, with an MIC value of 0.05 µM [139].

Two new carboxamides, vochysiamides A (**149**) and B (**150**), and a known metabolite, 2,5-dihydroxybenzyl alcohol (**151**), have been isolated from the endophytic fungus *Diporthe vochysiae* LGMF1583 obtained from the medicinal plant *Vochysia divergens.* Compound **150** displays a significant activity against the Gram-negative bacterium carbapenem-resistant *K. pneumoniae*, with an MIC value of 80 μg/mL [140].

Six compounds, 1-acetyl-β-carboline (**152**), indole-3-carbaldehyde (**153**), tryptophol (**154**), 3-(hydroxyacetyl)-indole (**155**), brevianamide F (**156**), and cyclo-(l-Pro-l-Phe) (**157**), have been obtained from the endophytic bacterium *Aeromicrobium ponti*, isolated from the medicinal plant *V. divergens.* These compounds display moderate antibacterial activities against methicillin-sensitive *S. aureus* and methicillin-resistant *S. aureus*, with inhibition zones of 10–18 mm and 8–15 mm, respectively [141].

Chloramphenicol (**158**) and *cyclo*-(l-tryptophanyl-l-prolyl) (**159**) have been obtained from *Streptomyces* sp. SUK 25, isolated from the root of *Zingiber spectabile*. These compounds demonstrated strong activities against methicillin-resistant *S. aureus* ATCC 43300, with MIC values of 8–16 mg/mL [142].

Taechowisan et al. [143] have isolated methyl 5-(hydroxymethyl)furan-2-carboxylate (**160**) and geldanamycin (**161**) from the endophytic bacterium *Streptomyces zerumbet* W14, obtained from the rhizome tissue of *Zingiber zerumbet* (L.) Smith. Compound **160** has shown activities against *S. aureus* ATCC 25923 and the methicillin-resistant *S. aureus* strain Sp6 (clinical isolate) with MIC and minimum bactericidal concentration values of 160 µg/mL and 16–64 µg/mL, respectively.

These numerous examples illustrate the power of natural endophyte-derived compounds against antibiotic-resistant bacteria. However, in the future, it will be appropriate to increase the number of studies aiming at the determination of their mechanisms of action and overcoming the antibiotic-resistance mechanisms. These studies will be crucial for advancing our knowledge and potentially harnessing these compounds more effectively in the ongoing battle against antibiotic-resistant strains.

## 5. Biosynthetic Gene Clusters of Secondary Metabolites of Endophytic Fungi

Natural compounds isolated from endophytes exhibit diverse biological activities, such as antimicrobial, antitumor, antioxidant, and immunomodulatory properties, making them valuable sources for the development of novel pharmaceuticals, agrochemicals, and other bioactive compounds [144,145]. Genome-mining-based strategies offer new insights to discover novel natural endophyte-derived alternative compounds compared to the conventional bioactivity-guided screening approach [146]. The biosynthesis of these bioactive secondary metabolites is regulated by specific gene clusters within the genomes of endophytic microorganisms. These gene clusters encode enzymes responsible for the synthesis, modification, and transport of the secondary metabolites. The coordinated expression of these genes leads to the production of complex chemical compounds with unique structures and activities [147]. One of the strategies that has been employed to investigate the gene clusters of endophyte microorganisms is the “antibiotics and secondary metabolite analysis shell” (antiSMASH). The antiSMASH employs a combination of computational algorithms and databases to analyze microbial genomes and predict the gene clusters associated with the secondary metabolites’ biosynthesis [148]. The identification of the gene clusters involved in the biosynthesis of bioactive secondary metabolites in endophytic microorganisms gives possibilities for the manipulation and optimization of the compounds’ production [149]. Therefore, in this section, we will explore the gene clusters of bioactive secondary metabolites in endophytic microorganisms and their importance in natural product discovery.

*Fusarium* sp. R1, an endophytic strain, has been isolated from *Rumex madaio* Makino, and the antiSMASH analysis has been performed for this endophyte. According to the analysis, the genome of the strain R1 possesses 37 biosynthetic gene clusters (BGCs) encoding 13 polyketide synthetases (PKSs), 3 terpene synthases (Ts), 1 hybrid indole + nonribosomal peptide synthetase (NRPS), 3 hybrid NRPS + PKSs, and 17 NRPSs [150].

Secondary metabolites of *Epicoccum latusicollum* HGUP191049, derived from *Rosa roxburghii* Tratt, have shown strong activities against Gram-positive (*S. aureus* and *B. subtilis*) and Gram-negative (*P. syringae* pv. *actinidiae*, *E. coli*, and *P. aeruginosa*) bacteria, with diameters of inhibition zones ranging from 15 to 20 mm. The antiSMASH analysis has identified 24 BGCs, including three distinct gene clusters that control the synthesis of the antibacterial substances oxyjavanicin, patulin, and squalestatin S1 [151].

The genome of the fungus *Calcarisporium arbuscula*, an endophyte of *Russulaceae*, has been studied with the antiSMASH, identifying 65 BGCs for secondary metabolites. Genes encoding PKSs and NRPSs have been found in 23 and 12 gene clusters, respectively. Furthermore, some gene clusters are predicted to control production of mycotoxins, such as aurovertins, aflatoxin, alternariol, destruxins, citrinin, and isoflavipucine [152].

Six compounds, brefeldin A, 7-dehydrobrefeldin A, brefeldin C, methyl tetradecanoate, anthraquinone ZSU-H85, and (3β,5α,6β,22E)-ergosta-7,22-diene-3,5,6-triol, have been isolated from the EtOAc extract of *D. alcacerensis* CT-6 obtained from the medicinal plant *Corydalis tomentella*. The antiSMASH analysis of *D. alcacerensis* CT-6 has identified 45 secondary metabolites’ BGCs. Of those, two BGCs in the regions of 8.2 and 12.4 have emerged as being responsible for the synthesis of three of these isolated compounds: brefeldin A, 7-dehydrobrefeldin A, and brefeldin C [146].

Whole-genome sequencing and gene annotation have been completed for the *Fusarium multiceps* BPAL1 strain isolated from *A. ilicifolius*. Out of 11,675 predicted genes, 33 BGCs have been found using the antiSMASH analysis [153].

An endophytic strain, *Alternaria* sp. SPS-2, has been isolated from *Edgeworthia chrysantha Lindl*., followed by its whole-genome sequencing and antiSMASH analysis identifying 22 secondary metabolites’ BGCs encoding 7 PKSs, 10 NRPSs, 4 terpenes, and 1 fungal ribosomally synthesized and post-translationally modified peptide. Of these BGCs, 8 have been previously known to be responsible for the production of equisetin, betaenones A–C, alternariol, dimethylcoprogen, and melanin; 14 BGCs are newly identified. LS–MS/MS– and GNPS–based studies have shown that this endophytic fungus has the potential to create bioactive secondary metabolites [154].

In a study conducted by Wang et al. [155], 27 PKSs, 12 NRPSs, 5 dimethylallyl tryptophan synthases, 4 putative PKS-like enzymes, 15 putative NRPS-like enzymes, 15 terpenoid synthases, 7 terpenoid cyclases, 7 fatty-acid synthases, and 5 PKS–NRPS hybrids have been determined by the antiSMASH analysis after the whole-genome sequencing of *Pestalotiopsis fici*, the endophyte obtained from branches of the tea plant *Camellia sinensis*. The genes for most of these key enzymes are dispersed throughout 74 BGCs.

It has been reported that the *Grammothele lineata* strain SDL-CO-2015–1 derived from *Corchorus olitorius* can produce the anticancer compound paclitaxel in cultures. Therefore, Ehsan and colleagues [156] have examined genome of *G. lineata* using various bioinformatic techniques to determine the relationship between the genome and the synthesis of the anticancer compound Taxol. The antiSMASH predicts 29 secondary metabolite BGCs encoding 1 NRPS, 6 T1PKSs, 12 terpenes, and 10 NRPS-like proteins in the genome of *G. lineata*. However, among the identified gene clusters, no single gene cluster is responsible for the synthesis of Taxol.

*Alternaria burnsii* NCIM 1409, isolated from the plant *Nothapodytes nimmoniana*, which produces another anticancer compound, camptothecin, has been subjected to whole-genome sequencing and the antiSMASH analysis, which have identified 25 BGCs in this fungus, none of which is responsible for the synthesis of camptothecin. Interestingly, 37 candidate genes involved in camptothecin production have been determined by comparative studies with similar fungi. There is no indication that these genes were transferred horizontally from the host plant to the endophyte, suggesting that camptothecin production in this fungus evolved independently [157].

In another study, whole-genome sequencing of the endophyte *Fusarium* sp. VM-40, derived from the medicinal plant *Vinca minor*, has been conducted. The genome size of the endophyte has been determined as being 40 Mb; the antiSMASH has predicted a total of 56 BGCs, including 25 previously known. Through combined genomic and metabolomic analyses, 30 secondary metabolites have been identified, most of which are enniatins—cyclic hexadepsipeptides generated by condensing three N-methyl-L amino acids and three D-α-hydroxy acids [158].

Four compounds—cochlioquinone B, cochlioquinone D, 8-hydroxy-6-methyl-9-oxo-*9H*-xanthene-1-carboxylic acid methyl ester, and isofusidienol A—have been isolated from *Helotiales* sp. BL73, an endophyte of the medicinal plant *Bergenia pacumbis*. The antiSMASH analysis of the endophytic genome has identified 77 BGCs, including the gene clusters encoding 26 type I PKSs, 3 type III PKSs, 25 NRPSs, 6 PKS/NRPS hybrids, 13 terpenes, and 4 indoles. Moreover, four terpene genes have been identified and codon-optimized for expression in *Streptomyces* spp. As a result, the recombinant strains could efficiently produce linalool and its oxidized form, the terpenoids often found in plants, and an as-yet-unidentified terpenoid [159].

According to results of the whole-genome analysis of *Ascomycete sp. F53* isolated from *Taxus yunnanensis*, 35 metabolite BGCs have been determined, one of which being responsible for the tandem of the polyketide synthase pathway and the azaphilone biosynthesis pathway. Chemical investigation of the fungus has identified a novel compound, lijiquinone 1, which shows cytotoxicity against human myeloma cells (IC_50_ = 129 µM) and antifungal activities against *C. albicans* and *Cryptococcus albidus*, with IC_50_ values of 79 µM and 141 µM, respectively [160].

In summary, the recent surge in research focused on deciphering BGCs within endophytic fungi has yielded significant advancements. This ongoing exploration has led to the unveiling of previously unknown gene clusters, identification of novel biologically active substances, and a deeper comprehension of the intricate relationship between plants and endophytes. Moreover, substantial efforts are being directed toward the cloning of putative genes responsible for synthesizing these biologically active secondary metabolites, with a subsequent emphasis on the synthesis of their anticipated compounds. Despite the growing enthusiasm surrounding endophytes, it is noteworthy that only a limited number of isolated endophytes have undergone comprehensive genome sequencing, coupled with a detailed examination of their biosynthetic gene clusters. The primary hindrance to broader genomic investigations lies in the formidable cost associated with whole-genome sequencing and annotation.

## 6. Conclusions

Despite advancements in scientific research, the development of new antibiotics has not kept pace with the rapid development of resistance by pathogenic bacteria in recent decades. Finding new antibiotics is one of the most important issues facing modern medicine. One of the most interesting directions in the quest to discover new antibacterial substances is the search among the microbiomes of eukaryotes, such as plants.

It is important to note that plants, unlike animals, rely on the innate immunity of each cell and not on mobile protective cells or the somatic adaptive immune system [161]. Antimicrobial plant compounds—phytoanticipins and phytoalexins—thus, do not affect endosymbionts. If the population of endosymbionts must adapt to phytoanticipins then in the case of phytoalexins, it is enough to simply be a good endosymbiont and not have a pathogenic effect on the plant. However, other potentially phytopathogenic microorganisms can also cause the synthesis of phytoalexins by plants, which will adversely affect not only the pathogen but also its endosymbionts. Therefore, the synthesis of antibacterial substances by endophytes, which will suppress the growth of potentially pathogenic populations without negatively affecting the plant, is highly expected. It is necessary to take into account that the phytoalexins of some plants can be phytoanticipins in others [162], which makes it possible to obtain a huge combinatorial variety of potentially effective antibacterial substances when analyzing endophytes from various plants.

Each year witnesses increases in the number and diversity of publications devoted to the isolation of antibacterial drugs from endophytes. Unlike actinomycetes, which currently yield fewer and fewer novel compounds, plant microbiomes still represent a largely untapped ‘treasure chest’ in terms of their antibacterial compounds. This review has provided a comprehensive analysis of 161 antibacterial compounds isolated from endophytes over the past decade, and this is definitely far from the limit.

As we have detailed above, the majority of the compounds isolated from endophytes exert their antibacterial effects by destroying bacterial cell walls, disrupting membrane permeability, or inhibiting crucial bacterial proteins and enzymes. Because the detailed mechanisms by which these compounds exert their antibacterial effects and because the effects on antibiotic-resistant strains are often unclear, further investigations are needed. So far, no generalizable chemotypes have emerged among the antibacterial substances isolated from endophytes. It also remains understudied how effectively these compounds penetrate the bacterial cell envelope, which urges more research [85].

Natural compounds from endophytic microorganisms present a promising avenue for discovering potential antibiotic candidates. We strongly believe that the search for new antibiotics among plant microbiomes is one of the most successful trends in the fight against bacterial infections.

## Figures and Tables

**Figure 1 antibiotics-13-00271-f001:**
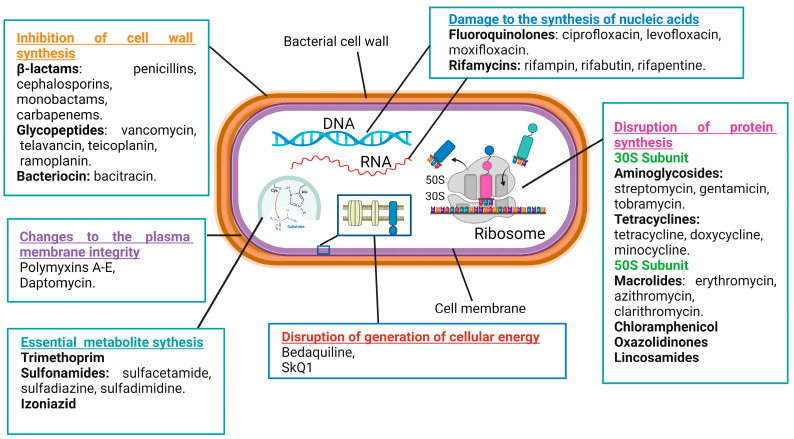
Mechanisms of action of antimicrobial drugs.

**Figure 2 antibiotics-13-00271-f002:**
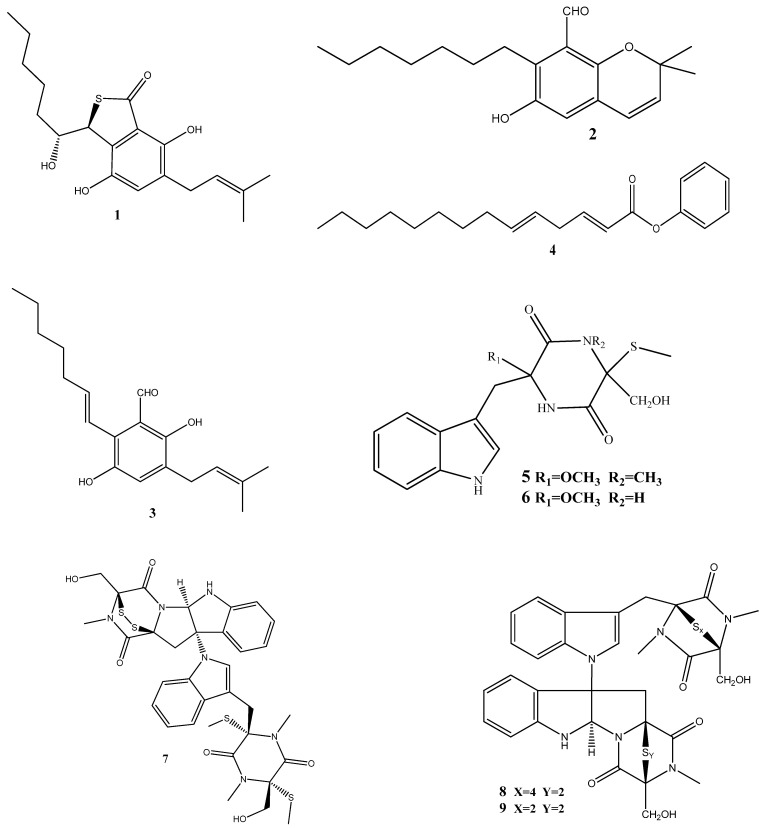
Chemical structures of endophyte-derived antibacterial compounds.

**Figure 3 antibiotics-13-00271-f003:**
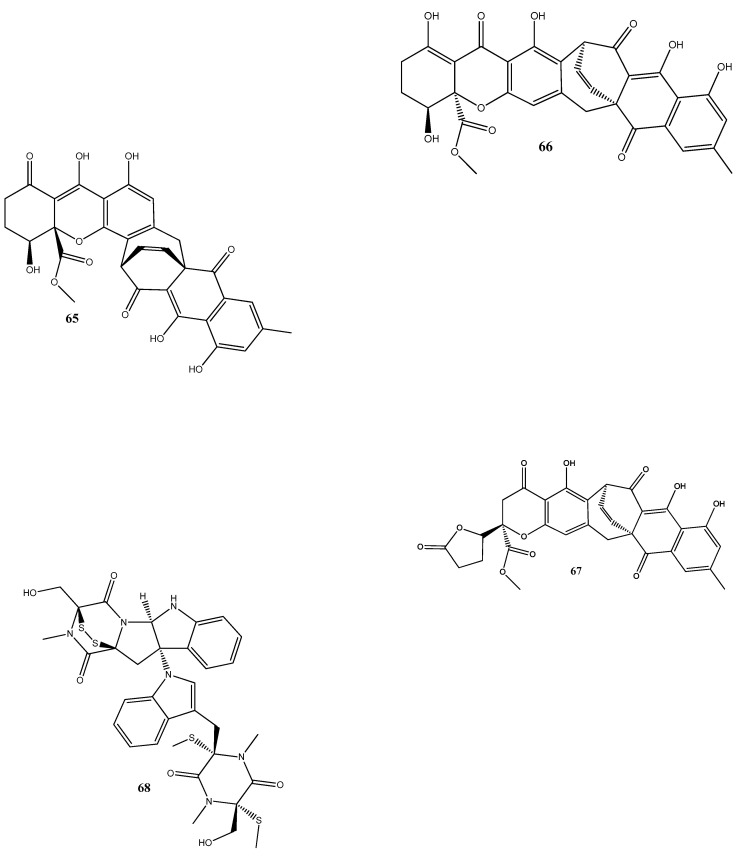
Chemical structures of endophyte-derived compounds active against antibiotic-resistant bacteria.

## Data Availability

Not applicable.

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
