# Peer review of "Antimicrobial Action Mechanisms of Natural Compounds Isolated from Endophytic Microorganisms"

_antibiotics, 2024, doi:10.3390/antibiotics13030271_

Round 1

Reviewer 1 Report

Comments and Suggestions for Authors

This review describes the mechanisms of actions of antimicrobial drugs and secondary metabolites isolated from endophytesantibacterial activities of the natural compounds derived from endophytes against antibiotic-resistant bacteria and biosynthetic gene clusters of endophytic fungi responsible for the synthesis of bioactive secondary metabolites. 

 I found the following after reading the manuscript.

Figures 8 and 9 do not show what X and Y mean

Figures 12 and 13 should be further apart

Figures 31 and 32 are the same, however, on line 275 are indicated as different compounds.

The text contained in lines 420 and 421 seems unfinished. 

Refrence 106, on lines 1048-1049Shimels Tikuye, Y. Review on Antibiotic ResistanceResistance MechanismsMethods of Detection and Its Controlling StrategiesBiomed J Sci & Tech Res 24 is incomplete 

Reference 117, on line 1071 is incomplete 

Reference 162, on line 1189 is incomplete 

Author Response

Thank you very much for your comprehensive analysis of the review. 

Reviewer 2 Report

Comments and Suggestions for Authors

The article Mechanisms of antimicrobial action of natural compounds isolated from endophytic microorganisms reviews the literature and describes the mechanism of antimicrobial action of natural products from fungi. In addition to its practical utility, the study also has substantial cognitive merit. In general, the paper is acceptable, and the work is qualified. The paper seems qualified to be published as a whole. However, the authors must clarify the methodology for collecting information by creating a new section.

Author Response

Thank you very much for your comprehensive analysis of the review and comments.

Reviewer 3 Report

Comments and Suggestions for Authors

In the past few decades, scientific communities have seen an exponential increase in antibiotic-resistant pathogens, and it is now essential to develop alternative/more effective new drugs against them. The paper has raised an important and valid question about the need to discover/develop new antimicrobial compounds either synthesized chemically or extracted from natural sources. In this review, the authors have tried to summarize all known antimicrobial compounds extracted from natural sources. I feel like the overall structure of the review is not very good. The review follows the repetitive presentation pattern, drug name, source, and very short description of the mode of action. This is a good, updated compilation of all naturally extracted antimicrobial compounds, but it failed to provide an in-depth mechanism of action of any discussed compound. The compounds from similar groups with overlapping modes of action can be discussed together rather than presenting them separately. Having them included in a table form with name, source, and targeted organism, will be very useful for the reader and it will also reduce the redundancy. Section 4 in the review does not align with the scope of the review.

I will suggest reducing the redundancy by analyzing the present understanding of various metabolites more comprehensively and concisely presenting them.

Author Response

(The authors gave the same response as above.)

Reviewer 4 Report

Comments and Suggestions for Authors

Comments for authors

 The manuscript entitled “Mechanisms of antimicrobial-action of natural compounds isolated from endophytic microorganisms” is clear and well-written. The content is complex with several supporting documents. Every point was supported by sources and actions on microorganisms. This version can be accepted after minor flaws will be corrected.

 1.      According to the format of the MDPI journal as follows:

 “All Figures, Schemes, and Tables should be inserted into the main text close to their first citation and must be numbered following their number of appearance (Figure 1, Scheme 1, Figure 2, Scheme 2, Table 1, etc.)”

 “All Figures, Schemes, and Tables should have a short explanatory title and caption”

 It does not meet the standard of the journal, kindly revise all parts.

 For example, Figures 1-3 should be inserted after line 181.

 2.      Line 168: Subsection 2.1 à Section 3?

 Good Luck!!!

Author Response

(The authors gave the same response as above.)

Reviewer 5 Report

Comments and Suggestions for Authors

The present review is focused on the “Mechanisms of antimicrobial-action of natural compounds isolated from endophytic microorganisms.” Infectious diseases are one of the big threats to human life for example COVID-19, in 2020 it showed the importance of the development of new medicine including antibiotics, vaccines, etc. In advanced scientific research, a major hurdle is the rapid development of resistance by pathogenic bacteria to antibiotics. To address the problem, there is a need to develop a new medicine to control infectious diseases. Particularly, this review described the mechanisms of actions of antimicrobial drugs and secondary metabolites isolated from endophytes, antibacterial activities of the natural compounds derived from endophytes against antibiotic-resistant bacteria, and biosynthetic gene clusters of endophytic fungi responsible for the synthesis of bioactive secondary metabolites. Natural active antibiotic compounds from endophytic microorganisms present a promising opportunity for discovering potential antibiotic candidates. This review would be useful in the search for new antibiotics among plant microbiomes against bacterial infections. This reviewer senses that the manuscript can be accepted for the publication considering following corrections.

1.     Page 1, line 44, In the introduction, please change the following sentence “It is predicted that in 2050, more than 10 million deaths will occur due to antibiotic-resistant bacteria.”

2.     Page 5, lines 187 to 190, the brackets of compound numbers are bold, remove it, e.g. chetoseminudin F (5) and G (6), and so on. Keep it the same format all over the manuscript.

3.     Page 6, line 263, the name of compound 29 is wrong correct it properly, also the structure is wrong in figure 2. Please check it and correct it.

4.     Structures drawn in Figures 2, 3, etc. all structures should be redrawn correctly. I would suggest using the ACS 1996 template in chem draw software.  For example, structures 5 and 6 shape of the ring is not correct, and compounds 14, and 26,37,39, need to be more attention.

5.     In Figure 3, compounds 124 to 140, are poorly drawn, it needs to be re-drawn. Please pay attention to the stereochemistry of atoms.

6.     In Figure 2, compound 11, check the sulfur linkages, it should be disulfide. Please check and correct it.

7.     Cited references are fine.

Author Response

(The authors gave the same response as above.)

Round 2

Reviewer 3 Report

Comments and Suggestions for Authors

The authors have addressed my concerns.